# Linking forward-pass dynamics in Transformers and real-time processing in humans

## Abstract

Modern AI models are increasingly being used as theoretical tools to study human cognition. One dominant approach is to evaluate whether human-derived measures are predicted by a model's output: that is, the end-product of a forward pass. However, recent advances in mechanistic interpretability have begun to reveal the internal processes that give rise to model outputs, raising the question of whether models might use human-like processing strategies. Here, we investigate the relationship between real-time processing in humans and layer-time dynamics of computation in Transformers, testing 20 open-source models in 6 domains. We first explore whether forward passes show mechanistic signatures of competitor interference, taking high-level inspiration from cognitive theories. We find that models indeed appear to initially favor a competing incorrect answer in the cases where we would expect decision conflict in humans. We then systematically test whether forward-pass dynamics predict signatures of processing in humans, above and beyond properties of the model's output probability distribution. We find that dynamic measures improve prediction of human processing measures relative to static final-layer measures. Moreover, across our experiments, larger models do not always show more human-like processing patterns. Our work suggests a new way of using AI models to study human cognition: not just as a black box mapping stimuli to responses, but potentially also as explicit processing models.

## 1 Introduction

One of the most exciting features of modern AI models—especially language models (LMs)—is their ability to capture fine-grained measures of human cognition (e.g., Frank & Goodman, 2025). For higher-order cognitive tasks, the prevalent comparison between humans and LMs is at the behavioral level. This approach typically involves using the LM to estimate the likelihoods of strings, which are linked to relevant behaviors on the human side, such as answer selections in a multiple-choice task. This "output-level" approach is often motivated on theoretical grounds. For example, probabilities derived from LMs enable systematic, large-scale testing of expectation-based theories of sentence processing (e.g., Levy, 2008; Smith & Levy, 2013; Huang et al., 2024; Michaelov et al., 2024b; Shain et al., 2024), as well as the relationship between string probability and grammaticality (e.g., Lau et al., 2017; Hu et al., 2025; Tjuatja et al., 2025).

At the same time, the field of mechanistic interpretability has begun to uncover the internal processes that support model outputs in reasoning or fact-retrieval tasks (e.g., Biran et al., 2024; Ghandeharioun et al., 2024; Merullo et al., 2024; Wendler et al., 2024; Lepori et al., 2025; Kim et al., 2025; Wiegreffe et al., 2025). A widespread assumption is that tasks that are "easy" for an LM can be solved in fewer layer-wise computation steps (Belrose et al., 2023; Baldock et al., 2021). Recently, Kuribayashi et al. (2025) found that predictions from earlier LM layers correspond more closely with "fast" measures of human sentence processing (such as gaze durations), while predictions from later layers correspond with "slow" signals (such as N400 event-related potentials). Their findings raise an open question: given an input stimulus, does the *information processing* involved in a forward pass of a model resemble the *cognitive processing* involved in producing a human's response? If so, this would suggest a new way of using ML models to generate insights into human processing.

Here, we use simple mechanistic analyses to investigate the relationship between layer-wise processing dynamics in Transformers (Vaswani et al., 2017) and real-time processing in humans (Figure 1).

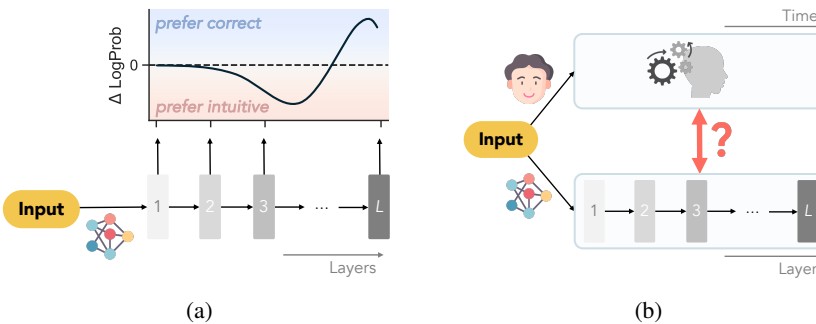

Figure 1: **Overview of our studies.** (a) Study 1: We explore whether forward passes show mechanistic signatures of competitor interference, first preferring a salient intuitive answer before preferring the correct answer. (b) Study 2: We systematically investigate the ability of dynamic measures derived from forward passes to predict indicators of processing load in humans.

We address three major research questions, listed below. In our first study, we take inspiration from cognitive science to provide high-level hypotheses about how computation might unfold in a forward pass. We explore whether forward passes of Transformers show **mechanistic signatures of competitor interference effects** inspired by dual-processing theories (e.g., Wason & Evans, 1974; Sloman, 1996; Evans, 2008; Kahneman, 2011). Building on prior exploratory work (Hu & Franke, 2024), we propose new measures for diagnosing delayed decision-making and two-stage processing in LMs and investigate cases where, at least for humans, a salient intuitive answer competes with the correct answer and may even be temporarily preferred before the correct answer "wins." In our second study, we then perform a quantitative, systematic study of whether these and other more general measures of processing difficulty from a forward pass **increase predictive accuracy** of processing-related aspects of human behavioral data. Finally, across both studies, we also investigate the **effect of model size**.

> **RQ 1:** *Do models show signs of **competitor interference effects**, with **delayed decision-making** and **two-stage processing**?*

> **RQ 2:** *Do measures characterizing **(a) competitor interference effects** or **(b) other aspects of processing difficulty** in models increase the accuracy of a (linear) model **predicting human processing load**, above and beyond static measures derived from model outputs?*

> **RQ 3:** *How does **model size** affect the similarity between model and human processing?*

We investigate these RQs across 20 open-source models and 6 domains, covering multiple modalities (text and vision) and human behavioral measures. To foreshadow our results, we find that LM forward passes show signs of competitor interference, with delayed decision-making and two-stage processing, for the items that have salient intuitive answers that compete with the ground-truth correct answer. Moreover, we find that measures of layer-time dynamics improve the ability to predict human processing indicators, above and beyond static measures derived from the final layer or an intermediate layer. We also find interactions between these trends and model size: larger models are not always most predictive of human processing, and mid-size LMs show the strongest signs of two-stage processing. Our results provide suggestive evidence that model processing and human processing may be facilitated or impeded by similar properties of an input stimulus. Furthermore, this apparent similarity seems to have emerged through general-purpose objectives such as next-token prediction or image recognition.

## 2 DOMAINS FOR EVALUATING COGNITIVE PROCESSING PATTERNS

In this section, we describe the domains used to evaluate cognitive processing patterns in our experiments (Table 1). A critical subset of the test items are expected to trigger two-stage processing, as discussed below. We refer to these items as belonging to the "Competitor" condition, since they

Table 1: Overview of domains investigated in our studies. "$n$AFC" = $n$-alternative forced choice.

| Domain | Human task | Modality | Data source | Studies |
|---|---|---|---|---|
| Capitals (recall) | Free text response | Text | Ours | 1 & 2 |
| Capitals (recognition) | 2AFC (button press) | Text | Ours | 1 & 2 |
| Animal categories | 2AFC (mousetracking) | Text | Kieslich et al. (2020) | 1 & 2 |
| Gender bias | NA | Text | Lepori et al. (2025) | 1 |
| Syllogisms | 2AFC | Text | Lampinen et al. (2024) | 1 & 2 |
| Object recognition | 16AFC | Vision | Geirhos et al. (2021) | 2 |

have a salient incorrect answer that intuitively "competes" with the ground-truth correct answer. Additional details about the human data are provided in Section A.1.

**Capitals (recall).**  We begin with a standard fact recall task: retrieving the capital city of a country or United States state (Merullo et al., 2024). We manually curated 62 items (22 countries, 40 states), each consisting of a political entity, the correct capital city, and an incorrect city in the entity. The Competitor condition contained 42 items where the incorrect answer is the most populous city within its political entity (e.g., Illinois/Springfield/Chicago), which may potentially trigger competitor interference effects via reasoning similar to availability (Tversky & Kahneman, 1973) or recognition heuristics (Goldstein & Gigerenzer, 2002). The NoCompetitor condition contained 20 items, where the incorrect answer is not expected to compete with the correct answer (e.g., France/Paris/Marseilles).

We collected data from 56 self-reported English native speakers from the US on Prolific. Due to resource constraints, we only collected data for the 42 Competitor items. On each trial, participants saw a question of the form "What is the capital of ¡ENTITY¿?" and freely typed their answer in a text box. Each keystroke and its timestamp was logged. We then considered 6 dependent variables (DVs) which may reflect human processing difficulty, including accuracy, response time (RT), and 3 measures derived from typing patterns: time of the first keypress after the last time the box was empty,[1] the ratio between the total number of keypresses and the length (in characters) of the final submitted answer, and the number of times the participant pressed the backspace key.

**Capitals (recognition).**  Using the same stimuli as above, we then tested factual knowledge of capital cities in a *recognition* (i.e., forced choice) setting, where both the correct and intuitive options are presented. We collected data from 41 human participants on Prolific. Again, we only collected data for the 42 Competitor items. On each trial, participants saw a question like "What is the capital of ¡ENTITY¿?" along with the correct and incorrect answers, and had to choose between the two answer options. Here, we only consider two DVs: accuracy and RT.

**Animal categories.**  Categorization of atypical exemplars (e.g., categorizing a whale as a mammal) can also induce processing difficulty. We used the stimuli and human data ($N$=108 participants) from Experiment 1 of Kieslich et al. (2020). The stimuli consist of 19 animals, paired with a correct category and an incorrect category. There are 6 items in the Competitor condition, where the animal is an *atypical* exemplar of the correct category (e.g., whale/mammal/fish), and 13 items in the NoCompetitor condition, where the animal is a *typical* exemplar (e.g., salmon/fish/mammal). Human participants saw the animal exemplar on the bottom center of the screen and the two category options in the top corners, and had to click on the correct category. Responses were collected via mousetracking, which provides fine-grained temporal and spatial information about cognitive processes during decision making (Spivey & Dale, 2006; Freeman, 2018; Stillman et al., 2018).

We considered 6 DVs, including accuracy, RT, and 4 measures derived from the mouse trajectories: AUC (area between the trajectory and a straight line from the start to the selected option), MAD (signed maximum deviation between the trajectory and a straight line from the start to the selected option), number of directional changes ("flips") on the $x$-axis, and the time of maximum acceleration.

**Gender bias.**  We then investigated decision conflict induced by contextual cues, adapting the gender bias stimuli from Lepori et al. (2025). The stimuli consist of 40 items, one for each of

---

[1]Due to technical difficulties, this particular DV was only recorded for 30 participants out of the total 56.

the professions from the WinoBias dataset (Zhao et al., 2018). Each item has two variants. In the Competitor condition, we create a contextual cue that violates the most prevalent gender associated with the profession,[2] and in the NoCompetitor condition, we use a cue that is consistent with the expected gender: e.g., "The carpenter is somebody's grandmother/grandfather. The carpenter is a _." The correct answer ("woman"/"man") is consistent with the cue, and the incorrect answer is inconsistent. There is no human data associated with this dataset, so we only use it in Study 1.

**Syllogisms.** Next, we explore a more challenging and practically relevant task: judging the logical validity of simple syllogistic arguments. We used the stimuli and human data from Lampinen et al. (2024). The stimuli consist of 192 simple syllogistic arguments (two premises and a candidate conclusion). The correct answer is either "valid" or "invalid", depending on the ground-truth logical validity of the conclusion, and the incorrect answer is either "invalid" or "valid". The Competitor condition includes the 48 stimuli which induce *content effects*: i.e., when the logical validity of the conclusion is inconsistent with people's prior beliefs, thus triggering competition from the intuitive but incorrect answer. The remaining 144 items are in the NoCompetitor condition. Here we only considered two human DVs: accuracy and RT.

**Object recognition.** Finally, we tested whether our approach would generalize to an entirely different modality: vision. We compared pre-trained vision Transformer models (ViTs) and humans on their out-of-distribution (OOD) object recognition abilities, using the stimuli and human data released by Geirhos et al. (2021). The stimuli include 17 datasets of OOD images. The images feature objects in 16 basic ImageNet categories (e.g., chair, dog), but with manipulations such as stylization or parametric degradations. Since the stimuli do not have paired "correct" and "intuitive but incorrect" answers, this domain is only analyzed in Study 2 (Section 3.2) as an exploratory test of generalization. Again, we only considered two human DVs: accuracy and RT.

## 3 INVESTIGATING COGNITIVE PROCESSING STRATEGIES IN A FORWARD PASS

We now turn to our main experiments investigating processing dynamics in Transformer models. The underlying intuition is that there can be different "amounts" of computation involved in mapping from an input to an output, even though the model contains a fixed number of layers (Dehghani et al., 2019; Belrose et al., 2023; Brinkmann et al., 2024; Lepori et al., 2025). One natural way to measure processing effort is to measure how the output token distribution changes throughout the layers of the model. For example, if a model is confident in the correct response to a stimulus in early layers, then that stimulus requires fairly little effort (Baldock et al., 2021). Indeed, this intuition has motivated "early exit" techniques, designed to reduce compute costs at inference time (Geva et al., 2022; Schuster et al., 2022).

**Preliminaries.** We begin by defining some basic notation. To analyze layer-wise dynamics, we read out a probability distribution over the next token from each intermediate hidden layer using the logit lens method (nostalgebraist, 2020). Let $L$ be the number of layers in the model; let $W_U \in \mathbb{R}^{d \times |\mathcal{V}|}$ be the unembedding matrix that maps from the final hidden layer to output logits, where $d$ is the hidden layer dimension and $\mathcal{V}$ is the model's vocabulary; and let NORM be the final layer-normalization mapping applied before submitting to the unembedding matrix. We apply the vocabulary projection $W_U$ to the hidden representation $\mathbf{h}_{\ell,i}$ at layer $\ell \in \{1, 2, \dots, L\}$ and token index $i$ (conditioned on all previous tokens $t_1, t_2, \dots, t_{i-1}$) to obtain a vector in $\mathbb{R}^{|\mathcal{V}|}$ of unnormalized logits over $\mathcal{V}$. Finally, we obtain the probability of a token $t_i$ at the $\ell^{\text{th}}$ hidden layer after normalizing the logits:

$$p(t_i \mid t_1, \dots, t_{i-1}; \mathbf{h}_{\ell,i}) = \text{SOFTMAX}\left(\text{NORM}(\mathbf{h}_{\ell,i})W_U\right)[\text{id}(t_i)] \tag{1}$$

When $\ell = L$, Equation (1) gives the final output probability of the model under normal decoding.

To measure *relative confidence* between two tokens conditioned on a context $\mathbf{c}$ at layer $\mathbf{h}_\ell$, we define

$$\Delta\text{LOGPROB}(\mathbf{h}_\ell, v_C, v_I; \mathbf{c}) = \log p(v_C \mid \mathbf{c}; \mathbf{h}_{\ell,|\mathbf{c}|+1}) - \log p(v_I \mid \mathbf{c}; \mathbf{h}_{\ell,|\mathbf{c}|+1}) \tag{2}$$

---

[2]These associations were taken from Zhao et al.'s original data, which are based on statistics from the US Department of Labor in 2017.

as the log-odds ratio of the correct over the incorrect answer, where $v_C$ and $v_I$ refer to the first tokens of the correct and incorrect answers, respectively. We will write $\Delta\text{LOGPROB}(\mathbf{h}_\ell)$ for simplicity when the context and answer options are clear.

**Models.** In the text-based domains, we evaluated 18 open-source, pretrained, autoregressive LMs, with 3 sizes for each of 6 families: GPT-2 (Radford et al., 2019), Llama-2 (Touvron et al., 2023), Llama-3.1 (Grattafiori et al., 2024), Gemma-2 (Gemma Team et al., 2024), OLMo-2 (Team OLMo et al., 2025), and Falcon-3 (Falcon-LLM Team, 2024). Collectively, the models range in size from 124M to 405B parameters. We evaluated base models, because fine-tuning can lead models to put probability mass on tokens that are irrelevant to the correct answer. In the vision domain, we evaluated two open-source vision Transformer models: ViT Small and ViT Base (Dosovitskiy et al., 2021). Additional details about the 20 tested models are provided in Section A.2, Table 3.

To address the potential brittleness of the logit lens method, we additionally used the tuned lens (Belrose et al., 2023) for the models in our evaluation suite that have a pretrained tuned lens. We found qualitatively similar results between the logit lens and tuned lens (Section D.2, Figure 7), so for simplicity we focus on the logit lens in the main text.

All data and code for our experiments are available at [ANONYMIZED URL].

### 3.1 STUDY 1: COMPETITOR INTERFERENCE EFFECTS IN LMS

Prior exploratory work (Hu & Franke, 2024) reported suggestive evidence for competitor interference effects in a small set of models using data from Hagendorff et al. (2023). Here, we introduce more precise measures of this phenomenon, and systematically apply them to novel and more diverse datasets with controlled manipulations of model families and sizes.

We define two measures that capture different aspects of competitor interference, based on the model's layer-wise vocabulary distributions. First, we propose a novel "change of mind" measure COM, which is designed to capture two-stage processing within a forward pass. COM measures the magnitude of the maximum preference for the intuitive answer relative to the final preference for the correct answer. Let $m$ be the relative confidence given by the layer that most favors the intuitive answer (over the correct answer); i.e., $m = \min_{\ell \in \{1,2,\ldots,L\}} \Delta\text{LOGPROB}(\mathbf{h}_\ell)$. We define COM as:

$$\text{COM} = \begin{cases} 0 & m \geq 0 \\ \min\{0, \Delta\text{LOGPROB}(\mathbf{h}_L)\} - m & m < 0 \end{cases} \tag{3}$$

COM is 0 when the intuitive answer is never preferred over the correct one (i.e., when $m \geq 0$), and is otherwise larger when there is a larger difference between $m$ (the strongest preference for the intuitive answer) and the log-odds at the final layer $\mathbf{h}_L$.

The second measure, TTD, captures the "time to decision": i.e., the time (in layers) at which the model begins consistently preferring the correct answer. It is defined as the last layer at which all subsequent log-odds are non-negative; i.e., $\text{TTD} = \max\{\ell^* \in \{1, 2, \ldots, L\} : \forall \ell \geq \ell^*, \Delta\text{LOGPROB}(\mathbf{h}_\ell) \geq 0\}$. If the model never favors the correct answer, $\text{TTD} = L$. To facilitate comparison across models, the layer is normalized by the number of layers $L$. This is similar to the "prediction depth" metric in the early exiting literature (Baldock et al., 2021; Belrose et al., 2023).

**Result # 1: LMs show signs of competitor interference.** We first ask whether models show mechanistic signs of competitor interference effects (RQ 1). Figure 2a illustrates the mean COM and TTD measures across conditions and domains. The results are broadly consistent with competitor interference: both measures are generally higher for the subset of items that involve salient intuitive answers, compared to the subset of items that only involve incorrect answers. The exception is COM in the reasoning-based domains (gender bias and syllogisms).

To qualitatively illustrate different processing strategies across conditions, Figure 2b shows $\Delta\text{LOGPROB}$ across layers for three example models in the capitals recall domain. We first focus on the Competitor condition (left facet). OLMo-2 13B, a mid-sized model, shows the signature pattern of two-stage processing: a preference for the intuitive answer in intermediate layers, followed a preference for the correct answer in later layers. In contrast, the smallest tested model, GPT-2, shows a consistent preference for the intuitive answer, and the largest tested model, Llama-3.1 405B, shows

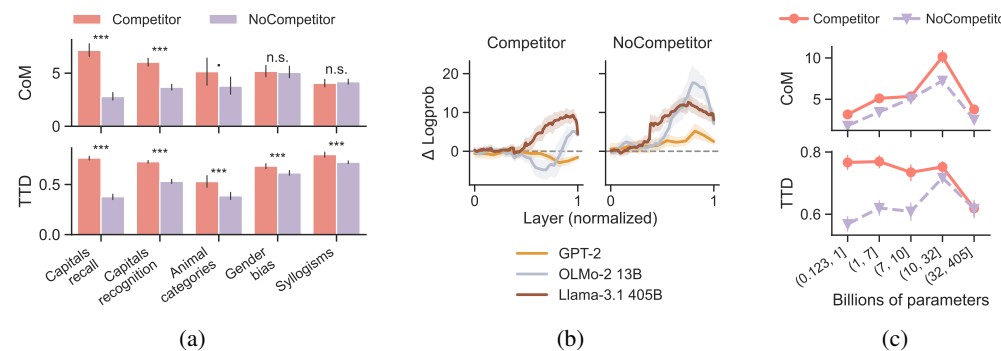

(a)                                        (b)                                        (c)

Figure 2: **Study 1 results.** (a) LMs generally show stronger signs of two-stage processing for the items with competing intuitive answers. Asterisks denote sig. $t$-tests comparing means across conditions within each domain. (b) ΔLOGPROB across layers for sample LMs in the capitals recall domain, illustrating different processing strategies. (c) Two-stage processing interacts with size.

a consistent preference for the correct answer. In the NoCompetitor condition (right), however, each of the models consistently prefers the correct answer across layers.

These patterns suggest interactions between model size and competitor interference. To investigate this quantitatively, we analyzed CoM and TTD across bins of model sizes, shown in Figure 2c. We find that mid-size models tend to show higher CoM, regardless of whether the stimulus invites competition from an intuitive answer. We also find that the difference between TTD for Competitor and NoCompetitor stimuli descreases as model scale increases, and that the largest models have lower TTD, regardless of condition.

### 3.2 STUDY 2: SYSTEMATIC COMPARISON OF HUMAN AND MODEL PROCESSING DYNAMICS

In Study 1, we found representational signatures of two-stage processing in LMs in the cases where we might expect them in humans. In humans, such differences in information processing are expected to manifest in different empirically measurable signals, such as complex speed–accuracy tradeoffs. This raises the question: does information from Transformers' forward passes also supply information about such fine-grained empirical indices of human processing? Study 2 therefore systematically explores a much larger set of measures, and their ability to quantitatively predict various empirical measures related to human processing.

#### 3.2.1 CANDIDATE METRICS

We consider several candidate model-derived measures for predicting human processing measures, summarized in Table 2. On top of CoM and TTD (see Section 3.1), the new measures are derived from five base metrics, and represent either *static* or *dynamic* measures of a model's forward-pass computation. Full details of how these metrics are computed are given in Section B.

Each base metric is a function $M: \mathcal{V}^* \times \mathbb{R}^d \times \cdots \to \mathbb{R}$ that maps a context $\mathbf{c} \in \mathcal{V}^*$, a hidden representation $\mathbf{h}_\ell$ from layer $\ell$, and possibly some other arguments onto a real number. The base metrics capture different aspects of the model's decision-making at a given layer. The **entropy** of the next-token distribution represents the model's uncertainty given the context $\mathbf{c}$. The **reciprocal rank** and the **log probability** of the first token of the correct answer represent the model's confidence in the correct answer. The **difference in log probability** ($\Delta \mathrm{LogProb}(\mathbf{h}_\ell)$, see (2)) captures the model's *relative* confidence in the correct answer. Finally, we define a "boosting" metric similar to the positional patching difference score of Kim et al. (2025), which characterizes how a particular layer contributes to $\Delta \mathrm{LogProb}(\mathbf{h}_\ell)$: i.e., does a layer increase or decrease the logit of the correct or incorrect answer?

Across the base metrics, we derive two broad types of quantities: **static** measures, which represent the application of a metric to a single layer; and **dynamic** measures, which aggregate the metric

Table 2: Model-derived measures used to predict human accuracy and processing measures in Study 2. "AUC+" = sum of positive values of $M$; "AUC-" = unsigned sum of negative values.

| Cognitive interpretation | Base metric $M$ | **Static** measures | **Dynamic** measures |
|---|---|---|---|
| Competitor interference | $\Delta$LOGPROB | N/A | COM, TTD |
| Uncertainty | Entropy | Final layer entropy, Middle layer entropy | AUC, Layer of max ↓ |
| Confidence | Reciprocal rank | Final layer rank, Middle layer rank | AUC, Layer of max ↑ |
| Confidence | LogProb | Final layer logprob, Middle layer logprob | AUC, Layer of max ↑ |
| Relative confidence | $\Delta$LOGPROB | Final layer $\Delta$LOGPROB, Middle layer $\Delta$LOGPROB | AUC+, AUC-, Layer of max ↑ |
| Boosting | Diffs projection | N/A | AUC+, AUC-, Layer of max value |

values across all layers $(M(\mathbf{c}, \mathbf{h}_1, \dots), M(\mathbf{c}, \mathbf{h}_2, \dots), \dots, M(\mathbf{c}, \mathbf{h}_L, \dots))$ into a single real number. We treat the static measures as a baseline, representing the kinds of measures typically used to link model computation and human behavior. Our main focus is the dynamic measures, which may supply additional information about real-time processing in humans.

The dynamic measures further fall into two subtypes of quantities: **area-under-the-curve (AUC)** quantities, which represent "influence over time" because they aggregate a metric across layers, e.g., $\sum_{\ell=1}^{L} M(\mathbf{c}, \mathbf{h}_\ell, \dots)$; or **argmax-delta** quantities, which represent the "time" at which the metric changes most quickly, operationalized as $\arg\max_{1 \le \ell \le L-1}\{M(\mathbf{c}, \mathbf{h}_{\ell+1}, \dots) - M(\mathbf{c}, \mathbf{h}_\ell, \dots)\}$. These subtypes mirror the two competitor interference metrics introduced in Section 3.1: both COM and AUC metrics measure the *magnitude* of some aspect of forward-pass dynamics, and both TTD and argmax-delta metrics are measurements of *time* (in layers).

### 3.2.2 ANALYSES

We perform the following analyses in each study (see Sections A.1 and C for additional details). First, to simply evaluate whether model "processing dynamics" predict human measures (e.g., accuracy, response time), we compute the coefficient of determination $R^2$ between item-level model-derived measures and human measures (averaged across individuals). However, even if a particular processing metric predicts substantial variance in a particular human DV, this might be because the metric is correlated with some static measure. Therefore, we additionally evaluate whether the dynamic measures *improve* prediction of human measures *beyond static measures*. For each LM and each human DV of interest,[3] we first fit a strong **baseline** mixed-effects regression model, which includes the interaction between all static predictors derived from the final layer, and random intercepts for participant (Equation (4)). For each dynamic measure, we then fit a new **critical** model that additionally includes the new independent variable (IV) (Equation (5)). We then compute the Bayes factor between the baseline and critical regression models.

```
DV ~ staticEntropy * staticRRank * staticLogProb * staticΔLogProb + (1|Subject)       (4)
DV ~ IV + staticEntropy * staticRRank * staticLogProb * staticΔLogProb + (1|Subject)   (5)
```

We repeated this analysis using static measures from an intermediate layer (i.e., the midpoint between the first and last layers) as the baseline predictors, to test whether the dynamic measures also improve upon static measures containing information about intermediate processing. We focus on the final-layer static baseline since these measures are most commonly used in model-human comparisons, and report results from the midpoint-layer static baseline in Section D.4, Figure 9.

### 3.2.3 RESULTS

**Result #2a: Competitor interference measures improve prediction of human processing measures.** We now return to RQ 2(a): do the measures of competitor interference (introduced in Section 3.1) predict human accuracy and processing load, *above and beyond static output measures*? We begin by analyzing the raw predictive power of the measures, illustrated by Figure 3a (top). Each point shows the proportion of variance of a particular human DV (e.g., accuracy, or # backspace

---

[3]Unless otherwise noted, we considered all trials for predicting human accuracy DVs, but restricted the analysis to trials where humans responded correctly for predicting human processing-related DVs.

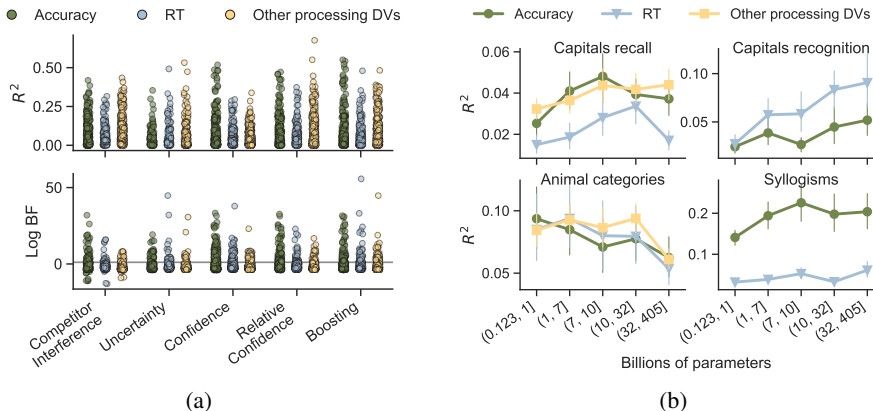

(a)  (b)

Figure 3: **Study 2 results for text domains.** (a) Top: $R^2$ achieved by model processing measures (x-axis) across groups of human DVs (hue). Bottom: Log Bayes Factor comparing critical to baseline regression models. Horizontal line $= \log(3)$.

presses) explained by a particular processing metric (e.g., CoM) for a given LM and domain. We see that the competitor interference measures (left group) are often strongly correlated with human DVs.

Next, we analyze the Bayes factors (BF) between the baseline and critical regression models (see Equations (4) and (5)), illustrated by Figure 3a (bottom). For the competitor interference measures (left group), many of the critical models improve upon the baseline models, achieving log BF > 0. These results demonstrate that competitor interference metrics substantially *improve* prediction of human DVs, above and beyond the output measures.

**Result #2b: Additional measures of processing dynamics improve prediction of human processing measures.** Having established the predictive power of the competitor interference measures, we turn to the broader set of processing measures introduced in Section 3.2.1, addressing RQ 2(b). We find similar results for the broader set of measures: there are many settings where the other processing measures explain substantial variance in human DVs (Figure 3a, top),[4] and many of the critical models improve upon the baseline models, achieving log BF > 0 (Figure 3a, bottom). These results also hold in the vision domain (Section D.3, Figure 8a) and with respect to the baseline models formed by static readouts from the midpoint layer (Section D.4, Figure 9).

Furthermore, it is not the case that all metrics are equally predictive of all DVs. Instead, we see some suggestive, potentially interpretable patterns in Figure 3a (top): on average, the confidence IVs predict more variance in accuracy DVs than processing-related DVs, whereas the uncertainty IVs predict more variance in processing-related DVs than accuracy. However, we do find a different pattern for the vision experiments, where accuracy is better predicted than RT overall, and uncertainty IVs predict more variance in accuracy than RT (Section D.3, Figure 8a).

**Result #3: Processing measures from larger models are not always better at predicting human processing measures.** Finally, we ask how model size affects the ability of a model's processing dynamics to predict human DVs (RQ 3). Figure 3b shows the mean $R^2$ values achieved across groups of DVs (hue) and quantiled bins of model parameter counts (x-axis) in the main LM experiments. It is *not* the case that larger models always explain the most variance in human DVs. Instead, we find that model size seems to interact with the task and the type of DV being explained. For capitals recognition and syllogisms, larger models tend to explain more variance in human accuracy and RTs. However, we find the opposite pattern for animal categorization, as larger models achieve lower $R^2$ on average for all groups of human DVs. For capitals recall, we observe diminishing returns to

---

[4]Many of these combinations result in $R^2$ values near 0. This is not necessarily problematic, since we don't expect *every* combination of models, model processing measures, and human measures to be strongly correlated.

model size for predicting human accuracy and RTs. Interestingly, these results seem to generalize prior findings for predictions of human reading times (e.g., Oh & Schuler, 2023; Shain et al., 2024) to a larger set of empirical measurements, and also extend prior observations by Hagendorff et al. (2023) to a more diverse class of models and measures.

## 4 GENERAL DISCUSSION

We found that "processing" metrics derived from forward pass dynamics predicted human task accuracy and processing measures, above and beyond static metrics derived from models' final-layer logits. This result held across multiple models, domains, human task modalities, and model input modalities. Our findings also suggest interesting interactions between model size and processing strategies: namely, larger models are not always most predictive of human processing, and mid-sized models are most likely to show competitor interference effects. An important direction for future work is to understand what properties of a model make it more or less human-like in its processing patterns, which mirrors prior studies of the relationship between model size, architecture, and training data for predicting measures of human language comprehension (e.g., Wilcox et al., 2020; Oh & Schuler, 2023; Michaelov et al., 2024a).

Our approach is similar to prior work comparing processing pipelines in models to hypothesized pipelines in cognitive or neural processing. For example, models optimized for language tasks seem to represent low-level linguistic information at earlier layers and build higher-level representations at later layers (Belinkov et al., 2017; Peters et al., 2018; Tenney et al., 2019; Belinkov et al., 2020; Lad et al., 2024), and models optimized for visual tasks such as object recognition or relational reasoning capture distinct stages of hierarchical processing in visual cortex (e.g., Yamins et al., 2014; Khaligh-Razavi & Kriegeskorte, 2014; Güçlü & van Gerven, 2015; Cichy et al., 2016; Nayebi et al., 2018; Lepori et al., 2024). We do not aim to link specific "time slices" of model/human processing, but instead compare high-level summary statistics of processing trajectories. While making such fine-grained sequential comparisons would be an interesting direction for future work, our approach also has the benefit of enabling comparison of models and humans on arbitrarily complex inputs, for which specific algorithms or brain pathways may be unclear.

From an AI perspective, our approach could be leveraged to characterize the processing difficulty associated with particular inputs for models. Prior work has shown that LMs are sensitive to auxiliary task demands at a behavioral level (Hu & Frank, 2024). However, what "counts" as a task demand is challenging to define *a priori*, without a deeper understanding of what induces processing difficulty. The processing metrics tested in our experiments could be candidate measures of "load" within models, which could then be applied to design evaluations with better construct validity, or more efficient techniques for early exiting to save test-time compute.

From a cognitive perspective, our work serves as a proof-of-concept, motivating future studies that explicitly investigate AI systems as models of human processing. Our results demonstrate that cognitively-inspired measures of model processing can predict human processing measures, suggesting some level of alignment between human and machine processing. Future work might seek to understand how modifications to architecture or training regimen mediate that alignment. For example, one could test whether modifications designed to induce dynamic or dual-stage strategies (e.g., van Gompel et al. 2001; Farmer et al. 2007; Van Bavel et al. 2012; cf. Hahn et al. 2022) lead to better predictions of human behavior using our metrics. Encouragingly, recent work in computer vision has attempted one such architectural modification (Iuzzolino et al., 2021), which has resulted in greater similarity to human subjects under time pressure (Subramanian et al., 2025). Other follow-up work could explicitly test the connection between forward-pass dynamics and reading times (building upon Kuribayashi et al. 2025), especially in settings where human syntactic processing is underestimated by surprisal (Wilcox et al., 2021; van Schijndel & Linzen, 2021; Arehalli et al., 2022).

Crucially, our work relies on intuitions from cognitive science in order to derive metrics that capture human behavior. An exciting avenue for future exploration would be attempting to do the inverse: characterizing aspects of model processing, demonstrating that those aspects predict human behavior, and then deriving novel insights into human cognition. Our work provides a necessary foundation for this pursuit, marking a first step toward using mechanistic analyses of AI models to understand processing in the human mind.

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

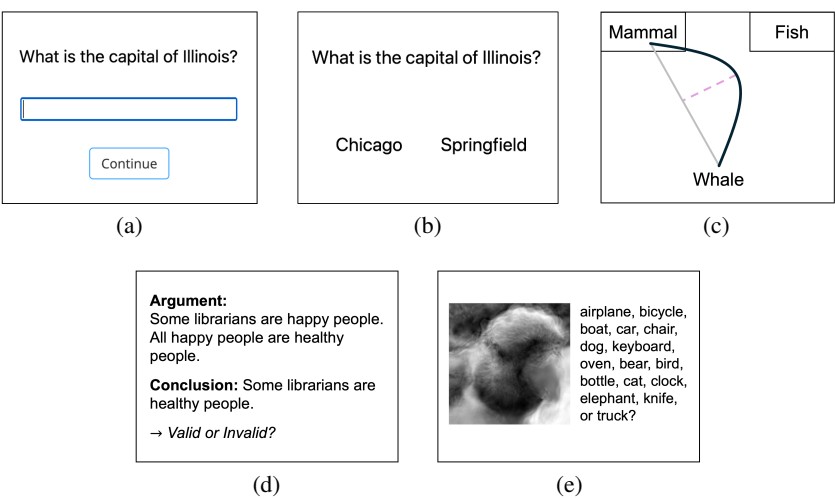

Figure 4: Illustration of human tasks analyzed in Study 2. (a) Recall (free response) of capital cities. (b) Recognition (forced-choice) of capital cities. (c) Categorization of typical and atypical animal exemplars via mouse movement (Kieslich et al., 2020). (d) Judgment of logical validity of syllogistic arguments (Lampinen et al., 2024). (e) Object recognition of out-of-distribution images (Geirhos et al., 2021).

## A DETAILS OF EXPERIMENTS

### A.1 HUMAN EXPERIMENTS

Below, we provide additional details about how the human data were collected, processed, and analyzed. The tasks are illustrated in Figure 4.

**Capitals recall.** We recruited 56 participants on Prolific, based in the United States with a self-reported native language of English. IRB approval was obtained under protocol [ANONYMIZED]. Participants were paid at a rate of $8/hr. Each participant saw each of the 42 Competitor items in randomized order, one on each trial. On each trial, participants saw a question of the form "What is the capital of ¡ENTITY¿?" and freely typed their answer in a text box (see Figure 4a). Each keystroke and its timestamp was logged. The text box had to be non-empty in order to advance to the next trial. Before the experiment, participants were asked to certify that they would not use any external tools such as search engines or generative AI to perform the study. They were also informed that their payment did not depend on the correctness of their answers, and were encouraged to guess if they were unsure of the answer.

For analyses, we excluded trials where the RT (between stimulus onset and submission of data) is more than 2 standard deviations away from the mean RT across all participants' trials. We also excluded trials (~5%) where the total number of keystrokes was less than the number of characters in the final answer. This occurred occasionally when people copied and pasted their answers, or when people's keystrokes were not recorded due to browser settings.

We consider 6 human behavioral DVs. First, we consider 2 measures of accuracy: **strict**, where a response is considered correct if it is an exact string match with the correct answer (after removing casing and whitespaces), and **lenient**, which allows for minor typos and spelling variations. We used GPT-4o to code responses for "lenient" accuracy (see Section A.3). Next, we consider 4 measures of processing load, consisting of **response time (RT)** and 3 measures derived from typing patterns: **time of the first keypress after the last time the box was empty**[5] (a measure of how long a participant "thinks" before typing their final answer), the ratio between the **total # of keypresses** and the length (in characters) of the final answer, and the **# of backspace presses** (a proxy for a participant's uncertainty).

---

[5]Due to technical difficulties, this particular DV was only recorded for 30 participants out of the total 56.

**Capitals recognition.**  We recruited 41 participants on Prolific, based in the United States with a self-reported native language of English.  IRB approval was obtained under protocol [ANONYMIZED]. Participants were paid at a rate of \$10.95/hr.  On each trial, participants saw a question like "What is the capital of ¡ENTITY¿?" and two answer options (correct and intuitive) beneath it (see Figure 4b). Their task was to press the "f" key to select the answer on the left-hand side of the screen, and the "j" key to select the answer on the right-hand side. Exactly half of the trials presented the correct answer on the left side, and the other half on the right side. As in the recall experiment (see above), each participant saw each of the 42 Competitor items, and the order of trials was randomized at runtime for each participant.  Again, we excluded trials with response times more than 2 standard deviations away from the mean.

Here, we only predict two human DVs of interest: accuracy and RT.

**Animal categories.**  We used the stimuli and human data from Experiment 1 of Kieslich et al. (2020). The stimuli consist of 19 animal exemplars, paired with a correct category and an incorrect category.  13 of the items are typical examplars of the correct category (e.g., "salmon"/"fish"), and the remaining 6 items are atypical examplars of the correct category (e.g., "whale"/"mammal"). While each item has an incorrect answer, the incorrect answer is only a salient *intuitive* answer for the atypical exemplars.

The human data consist of sequences of triples $(t, x, y)$: a timestamp $t$, and $x$- and $y$-coordinates of the participant's mouse.  There are 108 participants in total: 54 in the "click" group, where participants had to click on the region of the screen displaying their chosen category; and 54 in the "touch" group, where participants merely needed to move their mouse into that region of the screen. In the regression analyses, we included the group participants were assigned to as a fixed factor.

We considered 6 human DVs: **accuracy**, and 5 indicators of processing load or decision conflict. The processing-related DVs included **RT**, and 4 measures derived from the spatial mouse trajectories: **AUC** (area between the trajectory and a straight line from the start to the selected option), **MAD** (signed maximum deviation between the trajectory and a straight line from the start to the selected option), **# x flips** (number of directional changes on the $x$-axis), and **time of maximum acceleration**.

**Syllogisms.**  We used the stimuli and human data from Lampinen et al. (2024). Note that the data do not contain subject-level identifiers, but there are multiple variations of each item, so we include random intercepts for each item in the regression analyses. The stimuli consist of 192 items of the form $(e, \mathbf{a}^*, \mathbf{a}')$, where $e$ is the argument (two premises) and a candidate conclusion; $\mathbf{a}^*$ is "valid" or "invalid", depending on the ground truth of whether the conclusion logically follows from the argument; and $\mathbf{a}'$ is the incorrect label ("invalid" or "valid").  There are 96 "realistic" items: 48 where the logical validity of the argument is *consistent* with prior beliefs, and 48 where the logical validity is *inconsistent* with prior beliefs, thus inducing a content effect.  In addition, there are 96 "nonsense" items, which contain nonsense words in place of semantically contentful words (e.g., "Argument: Some pand are ing. All ing are phrite. Conclusion: Some pand are phrite.").

Here we only considered two human DVs: accuracy and RT.

**Object recognition.**  We compared pre-trained vision Transformer models (ViTs) and humans on their out-of-distribution (OOD) object recognition abilities, using the stimuli and human data released by Geirhos et al. (2021). The stimuli include 17 datasets of OOD images. The images feature objects in 16 basic ImageNet categories (e.g., chair, dog), but with manipulations such as stylization or parametric degradations. For the 12 parametric image degradation datasets, images are subject to different levels of degradation; e.g., different levels of uniform noise. We include a `condition` variable in our baseline and critical models for these 12 datasets to account for this. This factor was omitted for the 5 non-parametric manipulations.

Each item is of the form $(I, \mathbf{a}^*)$, where $I$ is a 224×224 image, and $\mathbf{a}^*$ is the correct category out of the 16 possible options.

Analogously to the logit lens for LMs, we derive intermediate model predictions by applying the final layernorm to the representation of the classification token after every intermediate layer, followed by the classification head. Since ViTs are encoder-only models (i.e., not autoregressive), there

Table 3: Overview of models evaluated in our experiments. (a) Language models. (b) Vision models.

(a)

| Model | HuggingFace ID | # params (B) | # layers | Vocab size | Training |
|---|---|---|---|---|---|
| GPT-2 | gpt2 | 0.124 | 12 | 50K | 40 GB |
| GPT-2 Med | gpt2-medium | 0.355 | 24 | 50K | 40 GB |
| GPT-2 XL | gpt2-xl | 1.5 | 48 | 50K | 40 GB |
| Llama-2 7B | meta-llama/Llama-2-7b-hf | 7 | 32 | 32K | 2T |
| Llama-2 13B | meta-llama/Llama-2-13b-hf | 13 | 40 | 32K | 2T |
| Llama-2 70B | meta-llama/Llama-2-70b-hf | 70 | 80 | 32K | 2T |
| Llama-3.1 8B | meta-llama/Llama-3.1-8B | 8 | 32 | 128K | 15T+ |
| Llama-3.1 70B | meta-llama/Llama-3.1-70B | 70 | 80 | 128K | 15T+ |
| Llama-3.1 405B | meta-llama/Llama-3.1-405B | 405 | 126 | 128K | 15T+ |
| Gemma-2 2B | google/gemma-2-2b | 2 | 26 | 256K | 2T |
| Gemma-2 9B | google/gemma-2-9b | 9 | 42 | 256K | 8T |
| Gemma-2 27B | google/gemma-2-27b | 27 | 46 | 256K | 13T |
| OLMo-2 7B | allenai/OLMo-2-1124-7B | 7 | 32 | 100K | 4T |
| OLMo-2 13B | allenai/OLMo-2-1124-13B | 13 | 40 | 100K | 5T |
| OLMo-2 32B | allenai/OLMo-2-0325-32B | 32 | 64 | 100K | 6T |
| Falcon-3 1B | tiiuae/Falcon3-1B-Base | 1 | 18 | 131K | 80 GT |
| Falcon-3 3B | tiiuae/Falcon3-3B-Base | 3 | 22 | 131K | 100 GT |
| Falcon-3 10B | tiiuae/Falcon3-10B-Base | 10 | 40 | 131K | 2 TT |

(b)

| Model | pytorch-image-models ID | # params (B) | # layers |
|---|---|---|---|
| ViT Small | vit-small-patch16-224 | 0.022 | 12 |
| ViT Base | vit-base-patch16-224 | 0.086 | 12 |

is less training pressure to build up classification decisions in a single residual stream, which is necessary for deriving our processing metrics. Nevertheless, Vilas et al. (2023) have found that class representations are built up across ViT layers.

Since the stimuli do not have paired "incorrect" answers, we only measure uncertainty (i.e., entropy) and confidence in the correct answer (i.e., the reciprocal rank or log probability of the correct image category). Again, we only considered two human DVs: accuracy and RT.

## A.2 MODEL EVALUATION

We evaluated models using the nnsight package (Fiotto-Kaufman et al., 2025). Table 3 provides more details about the models evaluated in our experiments. Inference was run on NVIDIA A100-40GB and H100 80GB GPUs during April–May 2025.

Below, we provide the full input for evaluating LMs in each domain.

**Capitals recall.** For each item, we measured model predictions conditioned on a context like "The capital of Illinois is".

**Capitals recognition.** For each item $(e, \mathbf{a}^*, \mathbf{a}')$, we constructed two prefix contexts, $\mathbf{c}_1$ and $\mathbf{c}_2$:

$$\mathbf{c}_1 = \text{"The capital of } e \text{ is either } \mathbf{a}^* \text{ or } \mathbf{a}'. \text{ In fact, the capital of } e \text{ is"} \qquad (6)$$

$$\mathbf{c}_2 = \text{"The capital of } e \text{ is either } \mathbf{a}' \text{ or } \mathbf{a}^*. \text{ In fact, the capital of } e \text{ is"} \qquad (7)$$

We then average all relevant metrics across these two orderings.

**Animal categories.** We measured each LM's responses conditioned on the prefix $\mathbf{c}$ = "A $e$ is a type of" (or "An $e$ is a type of", depending on the first letter of the animal name).

We translated the materials from Kieslich et al. (2020) from German into English for evaluating LMs.

**Syllogisms.** We measured LM responses conditioned on the following prefix, which is slightly modified from the prompt used by Lampinen et al. (2024):

```
In this task, you will have to answer a series of questions. You will have to choose
the best answer to complete a sentence, paragraph, or question. Please answer them to the
best of your ability.

Please assume that the first two sentences in the argument are true. Determine whether
the argument is valid or invalid, that is, whether the conclusion follows from the first
two sentences:
Argument: <ARGUMENT>
Conclusion: <CONCLUSION>
Answer: The argument is
```

### A.3 ANNOTATING RESPONSES IN CAPITALS RECALL EXPERIMENT (STUDY 1A)

We used the following prompt to query GPT-4o (April 2025) through the OpenAI API for labeling the responses from the capitals recall experiment.

```
People were asked to name the capital city of various countries and US states,
and your job is to label their responses. The possible labels are "Correct"
(the correct capital), "Intuitive" (the intuitive capital), "Alternate city"
(another city in the entity), "Not sure" (expressions of uncertainty), or "Other".

It's ok if the responses include minor typos or spelling variations.
Please provide the label that best describes the response.

Here are some examples.

Entity: Illinois
Correct: Springfield
Intuitive: Chicago
Response: "springfield"
Label: Correct

Entity: Pennsylvania
Correct: Harrisburg
Intuitive: Philadelphia
Response: "Philladelphia"
Label: Intuitive

Entity: Morrocco
Correct: Rabat
Intuitive: Marrakesh
Response: "Casablanca"
Label: Alternate city

Entity: Maryland
Correct: Annapolis
Intuitive: Baltimore
Response: "idk"
Label: Not sure

Entity: Canada
Correct: Ottawa
Intuitive: Toronto
Response: "Canada"
Label: Other

Now, here is a new item for you to label.

Entity: {entity}
Correct: {correct}
Intuitive: {intuitive}
```

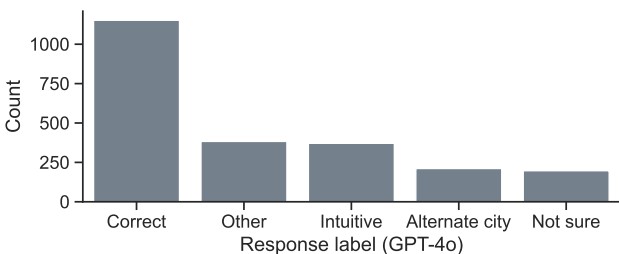

Figure 5: Distribution of labels assigned by GPT-4o to free responses in Study 1a (capitals recall).

```
Response: "{response}"

How would you label this response? Only respond with "Correct", "Intuitive",
"Alternate city", "Not sure", or "Other" as your answer.
```

We included examples with minor typos (e.g., "Philladelphia") and capitalization variations (e.g., "springfield") in the prompt.

For each trial, the actual entity, correct answer, intuitive answer, and human response were substituted in the `{entity}`, `{correct}`, `{intuitive}`, and `{response}` placeholders, respectively.

The resulting distribution of labels is shown in Figure 5.

## B DETAILS OF MODEL-DERIVED METRICS

### B.1 UNCERTAINTY

We first consider the model's "uncertainty" at the moment of decision-making, given by the entropy of the next-token distribution given context $\mathbf{c}$:

$$\text{ENTROPY}(\mathbf{c}; \mathbf{h}) = -\sum_{v \in \mathcal{V}} p(v|\mathbf{c}; \mathbf{h}) \log p(v|\mathbf{c}; \mathbf{h}) \tag{8}$$

Note that this measure depends only on the context, and not any particular answer option.

The output measure of uncertainty is given by the entropy of the output distribution at the final layer (**EntropyFinal**). The processing measures of uncertainty are given by the summed entropy across layers (**EntropyAUC**), and the layer index $\ell^* \in [1, L-1]$ of the largest *decrease* in entropy (**EntropyLayer**).

### B.2 CONFIDENCE IN CORRECT ANSWER

Next, we consider the model's degree of "confidence" in the correct answer. We consider two measures of confidence: (1) the log probability and (2) the reciprocal rank of the first token of the correct answer, both conditioned on the context $\mathbf{c}$.

Let $\mathbf{c}$ be the context (item) that the model is conditioned on, and let $\mathbf{a} = [a_1, a_2, \ldots, a_{|\mathbf{a}|}]$ be the answer string that we want to score, consisting of $|\mathbf{a}|$ tokens. We compute the log probability of the first token $a_1$ by applying log softmax to the logits (i.e., log on top of Equation (1)).

We also analyze the reciprocal rank of the first token $a_1$ of the answer $\mathbf{a}$ within the logits given by a particular layer:

$$\text{RANK}^{-1}(\mathbf{a}, \mathbf{c}; \mathbf{h}) = \frac{1}{\text{Rank}(\text{id}(a_1), \text{LOGITS}(\mathbf{h}|\mathbf{c}))} \tag{9}$$

where $\text{id}(a_1)$ gives the token index of token $a_1$. If $a_1$ is the top-ranked token, then this value will be 1, and if it is the bottom-ranked token, then this value will be $\frac{1}{|\mathcal{V}|}$.

The output measures of confidence are the reciprocal rank and log probability of the correct answer at the final layer (**RRankFinal** and **LogprobFinal**, respectively). There are four corresponding

processing measures of confidence: the area under the curves (**RRankAUC**[6] and **LogprobAUC**), as well as the layer indices of largest increase (**RRankLayer** and **LogprobLayer**).

## B.3 CONFIDENCE IN CORRECT ANSWER, RELATIVE TO INTUITIVE ANSWER

Next, we consider the model's confidence in the correct answer, *relative to* an alternate answer. In our experiments, this alternate answer is an intuitively salient (but incorrect) answer. We measure the relative confidence at a given layer $\mathbf{h}$ as the difference in log probability between the correct answer $\mathbf{a}^*$ and intuitive answer $\mathbf{a}'$, conditioned on the context $\mathbf{c}$:

$$\Delta\text{LOGPROB}(\mathbf{a}^*, \mathbf{a}'|\mathbf{c}; \mathbf{h}) = \text{LOGPROB}(\mathbf{a}^*|\mathbf{c}; \mathbf{h}) - \text{LOGPROB}(\mathbf{a}'|\mathbf{c}; \mathbf{h}) \tag{10}$$

The output measure of relative confidence is the log probability difference at the final layer ($\Delta$**LogprobFinal**). For processing measures, we obtained three metrics based on the curve formed by the logprob differences. First, we computed two AUC-based metrics: the area *above* 0 ($\Delta$**LogprobAUC+**), which measures the amount of "time" and confidence with which the model preferred the correct answer $\mathbf{a}^*$, and the area *below* 0 ($\Delta$**LogprobAUC-**), which measures the amount of "time" and confidence with which the model preferred the intuitive answer $\mathbf{a}'$. Note that these two quantities are not redundant—they could both be high, both be low, or one could be high while the other is low. Finally, we computed the layer at which we see the largest increase in the log probability difference between the correct and intuitive answers ($\Delta$**LogprobLayer**).

## B.4 BOOSTING OF CORRECT ANSWER, RELATIVE TO INTUITIVE ANSWER

We consider signatures of a model "boosting" the correct answer, relative to an alternate intuitive answer. Measures derived from the output layer alone do not give information about "boosting" dynamics, so we only consider processing measures in this case.

In the residual stream view of a transformer (Elhage et al., 2021), the effect of any layer, $\mathbf{h}$, in a Transformer model can be summarized by the delta between the residual stream before and after that layer, $\Delta\mathbf{h}$. Notably, $\Delta\mathbf{h}$ is simply another vector of the same dimensionality of $\mathbf{h}$, so one can project it into the vocabulary space using logit lens. We wish to quantify how different layers promote a correct answer over an intuitive answer. To do so, we computed the "logit difference" for each item with correct answer $\mathbf{a}^*$, intuitive answer $\mathbf{a}'$, and context $\mathbf{c}$

$$\Delta\text{LOGIT}(\mathbf{a}^*, \mathbf{a}'|\mathbf{c}; \Delta\mathbf{h}) = \text{LOGIT}(a_1^*|\mathbf{c}; \Delta\mathbf{h}) - \text{LOGIT}(a_1'|\mathbf{c}; \Delta\mathbf{h}) \tag{11}$$

and the "term difference"

$$\Delta\text{TERM}(\mathbf{a}^*, \mathbf{a}'|\mathbf{c}; \Delta\mathbf{h}) = \left|\text{LOGIT}(a_1^*|\mathbf{c}; \Delta\mathbf{h})\right| - \left|\text{LOGIT}(a_1'|\mathbf{c}; \Delta\mathbf{h})\right| \tag{12}$$

between the first tokens of the correct and intuitive answer options. This gives us a tuple for each layer and item that describes (i) whether or not $\mathbf{h}$ increases the probability of generating $\mathbf{a}^*$ over $\mathbf{a}'$ and (ii) whether or not $\mathbf{h}$ primarily changes the log probability of $\mathbf{a}^*$ or the log probability $\mathbf{a}'$.

In this space, the direction of the $\langle 1, 1 \rangle$ vector can be interpreted as *boosting* the correct answer relative to the intuitive answer — the layer is increasing the probability of generating $\mathbf{a}^*$ over $\mathbf{a}'$ by increasing the log probability of $\mathbf{a}^*$ (rather than decreasing the log probability of $\mathbf{a}'$). Similarly, the direction of the $\langle -1, -1 \rangle$ vector can be interpreted as *boosting* the intuitive answer.[7]

For a given layer and item, we then compute the scalar projection of $\langle \Delta\text{TERM}(\mathbf{a}^*, \mathbf{a}'|\mathbf{c}; \Delta\mathbf{h}), \Delta\text{LOGIT}(\mathbf{a}^*, \mathbf{a}'|\mathbf{c}; \Delta\mathbf{h}) \rangle$ onto the $\langle 1, 1 \rangle$ vector. For simplicity, we write this quantity as $S(\mathbf{a}^*, \mathbf{a}'|\mathbf{c}; \Delta\mathbf{h})$:

$$S(\mathbf{a}^*, \mathbf{a}'|\mathbf{c}; \Delta\mathbf{h}) = \frac{\langle \Delta\text{TERM}(\mathbf{a}^*, \mathbf{a}'|\mathbf{c}; \Delta\mathbf{h}), \Delta\text{LOGIT}(\mathbf{a}^*, \mathbf{a}'|\mathbf{c}; \Delta\mathbf{h}) \rangle \cdot \langle 1, 1 \rangle}{\|\langle 1, 1 \rangle\|} \tag{13}$$

---

[6]Note that to compute **RRankAUC** we compute the area between the layer-wise $\text{RANK}^{-1}$ curve and the lowest possible reciprocal rank $\frac{1}{|\mathcal{V}|}$ (which is extremely small in practice).

[7]One can also interpret the direction of the $\langle -1, 1 \rangle$ vector as *suppressing* the intuitive answer and the $\langle 1, -1 \rangle$ as *suppressing* the correct answer. However, we do not observe these types of layers empirically.

We used the layer-wise scalar projection curve to derive three processing measures: the area *above* 0, which represents time and "strength" of boosting the correct answer (**BoostAUC+**); the area *below* 0, which represents time and "strength" of boosting the intuitive answer (**BoostAUC-**), and **Boost-Layer**, the layer with the largest scalar projection. Note that here we are looking at the maximum *value* instead of the maximum *change*, as in the other metric groups, since the projection is already a measure of change (i.e., boosting).

## C    DETAILS OF REGRESSION ANALYSES

To perform the model comparisons for each study, we fit (generalized) linear mixed-effects models in R (version 4.3.2) using the `lme4` package. All predictors were centered and scaled. In some cases, a predictor had only one unique value across all items (e.g., the rank of the correct answer always being 1), in which case the predictor was dropped from the analysis for that particular model and task.

We fit standard linear models using maximum likelihood for all DVs except for binary variables (accuracy) and count variables (number of backspaces, number of x-position flips), in which case we fit generalized linear mixed-effects models using `family=binomial(link='logit')` and `family='poisson'`, respectively. Time-based DVs (such as RT or max acceleration time) were log-transformed.

Log Bayes factors were computed on top of fitted `lme4` models using the `bayestestR` package.

## D    ADDITIONAL EXPERIMENTAL RESULTS

### D.1    TASK ACCURACY

Figure 6 shows the overall accuracy achieved by each model and humans on each task. For text-based tasks, model accuracy is defined as whether the model assigned higher total probability to the correct answer option (i.e., summed log probability across tokens) than the incorrect answer option. For vision-based tasks, model accuracy is defined as whether the model assigned highest probability to the correct image category (out of the 16 options). This notion of accuracy is evaluated in the "normal" way; i.e., at the final layer only.

### D.2    COMPARISON OF LOGIT LENS AND TUNED LENS

Figure 7 shows the distribution of $R^2$ values (predicting human DVs) achieved under logit lens and the tuned lens (Belrose et al., 2023), for the models in our experiments for which there exists a pre-trained tuned lens. The distributions are largely similar, so we focus on results from the logit lens in the main text.

### D.3    STUDY 2: RESULTS FROM VISION DOMAIN

Figure 8 shows the Study 2 results from the object recognition datasets.

### D.4    STUDY 2: LOG BAYES FACTORS WITH RESPECT TO MIDPOINT-LAYER BASELINE

Figure 9 shows the distribution of log Bayes Factors between critical and baseline regression models, with the static baseline measures taken from the midpoint layer of each model.

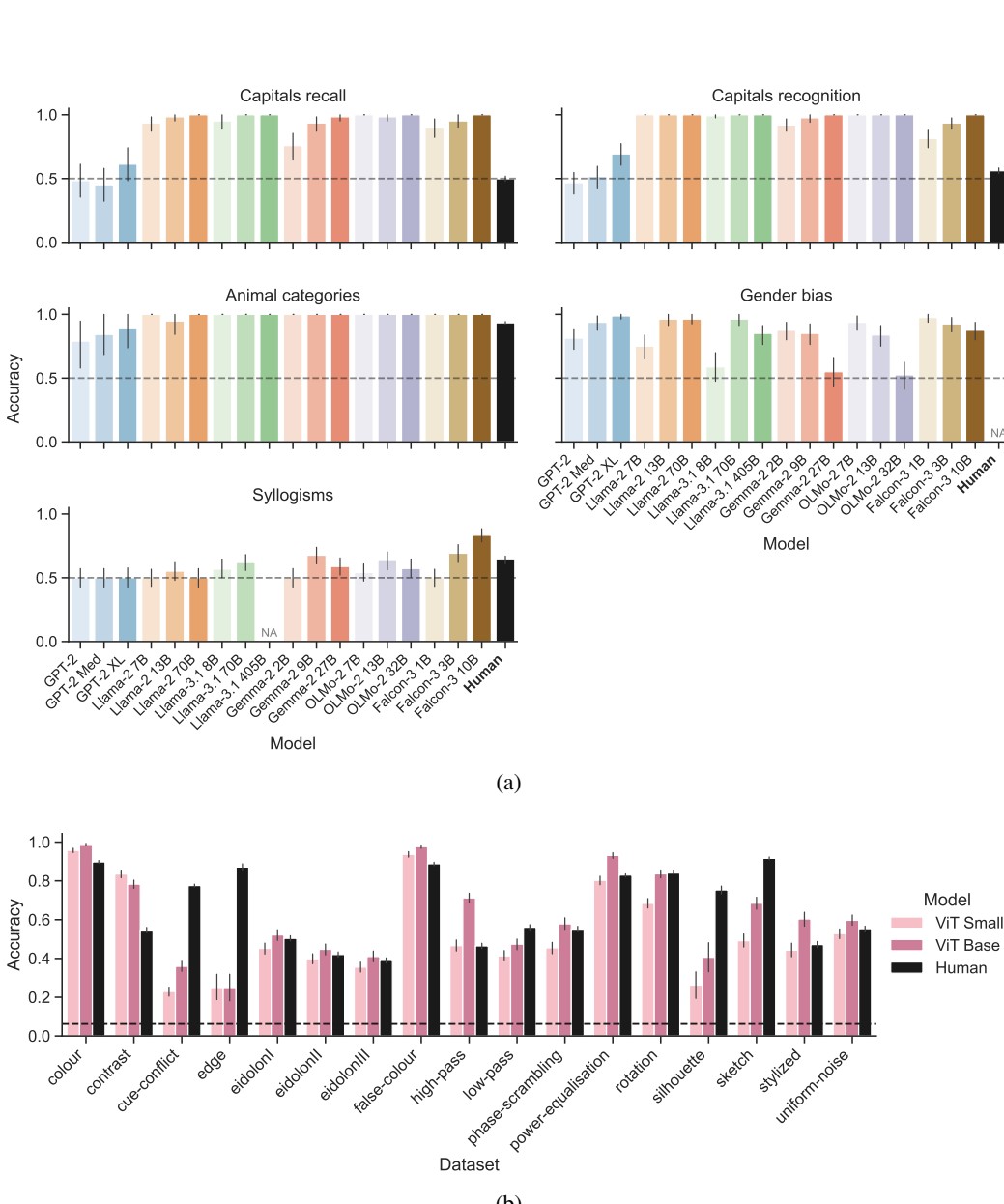

(a)

(b)

Figure 6: Accuracy achieved by models and humans in each domain. (a) Text-based domains. For capitals recall, we show the "lenient" definition of accuracy for humans (labeled as correct by GPT-4o, which allows for minor typos). Note that we only have human data for the Competitor items in the capitals domains, while the model accuracy is shown over both the Competitor and NoCompetitor conditions. We do not have data from Llama-3.1 405B on syllogisms, nor data from humans on gender bias. (b) Vision-based tasks (Study 4).

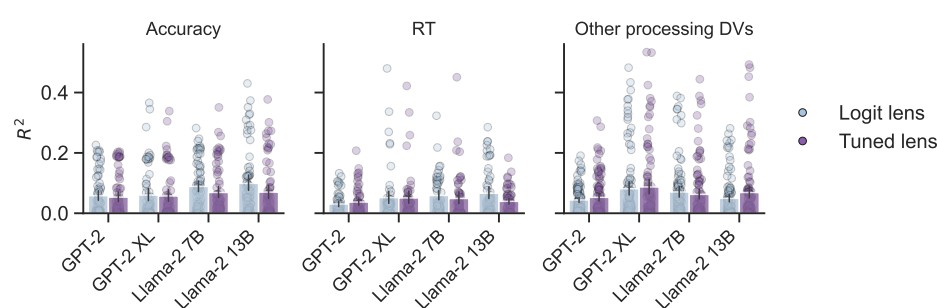

Figure 7: Distribution of $R^2$ values achieved under logit lens and tuned lens. Bars denote means.

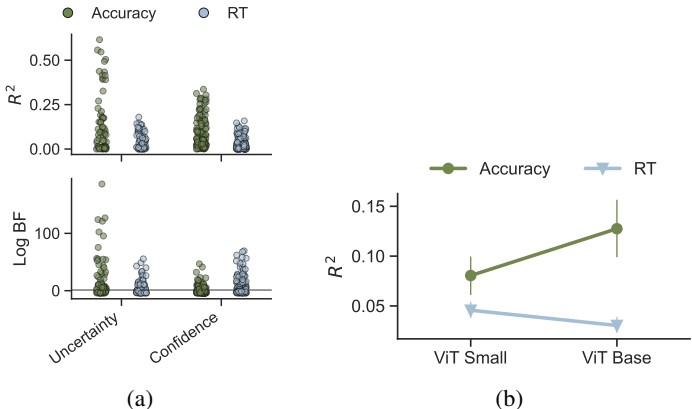

(a)                                          (b)

Figure 8: **Study 2 results for vision domain.** (a) Top: $R^2$ achieved by model processing IVs across groups of human DVs. Bottom: Log Bayes Factor comparing critical to baseline regression models (final-layer). Horizontal line denotes $\log(3)$. (b) Mean $R^2$ across models.

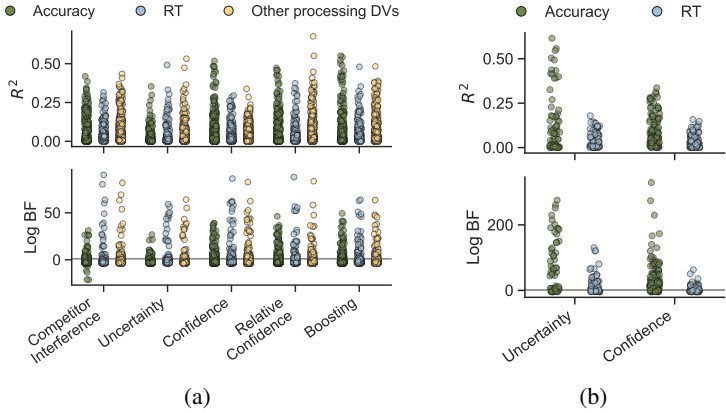

(a)                                          (b)

Figure 9: **Study 2 results, relative to midpoint-layer baseline.** Log Bayes Factor (bottom facets) comparing critical to baseline regression models, where the baseline is formed by static readouts from the midpoint layer. Horizontal line = $\log(3)$. Note that the $R^2$ data (top facets) does not depend on baseline measures, and is identical to the $R^2$ data shown in Figure 3a and Figure 8a (with small visual differences due to randomness in the jitter), and is shown again here for visual comparison to the log BF results. (a) Text domains. (b) Vision domains.

