# OpenReview forum: "Linking forward-pass dynamics in Transformers and real-time processing in humans"
_ICLR.cc/2026/Conference — ICLR 2026 Conference Withdrawn Submission_

### Official Review · Reviewer_SJiH · 2025-10-24

**Soundness:** 3
**Presentation:** 3
**Contribution:** 3
**Rating:** 6
**Confidence:** 5

**Summary:**

The authors investigate the relationship between human behavior and layer-time dynamics of computation in Transformer models. They investigate 20 LLMs across 6 different domains involving competitor interference. They find that different dependent variables can be predicted from measures derived from LLM activities. Dynamic measures did improve prediction of human processing measures relative to static final-layer measures and larger models did not always show more human-like processing patterns.

**Strengths:**

This is a strong paper that I enjoyed reading overall. The general premise of using AI models not just as a black box mapping but as explicit processing models has a lot of potential in cognitive science -- given that we would establish that this were possible.

In general the paper has all ingredients of a high-quality work: the set of experiments is interesting and diverse, multiple models are considered, and results are compared to actual human data. I particularly liked the fact that the authors focused on base models and 100% agree with them that this is a good design choice.

The paper was easy to read and follow, it builds on prior work but extends it in interesting and meaningful ways, and the plots are (for the most part) very intuitive.

**Weaknesses:**

Figure 3a was basically impossible to process for me visually. From what is visually presented, I couldn't draw any conclusions. The text also didn't help much as it only provided a qualitative description. Since this is one of the main results of the paper, this could definetly be improved. Plotting mean values (or something like a violin plot) would already help.

One piece of information that was missing: what is the accuracy of these models on the six tasks? Is it close to ceiling for all of them? That is very basic information but it would be good to know.

Minor:
* I found terms "Study 1" and "Study 2" a bit weird in this context. Personally, I would use other terms.
* RQ (on page 1) not defined.

**Questions:**

Something in the definition of TTD does not seem to add up: "the time (in layers) at which the model begins consistently preferring the correct answer" and "the last layer at which all subsequent log-odds are non-negative". Shouldn't the second sentence be "the first layer at which all subsequent log-odds are non-negative"?

Figure 3b: The R^2 values seem rather low in this plot. Can a noise ceiling be computed here? I would guess so, since there are answers from multiple participants per question.

I would be happy to increase my score to 8 if the two open questions are answered, and the readability of Figure 3a is improved.

---

### Official Review · Reviewer_1v2G · 2025-10-30

**Soundness:** 3
**Presentation:** 2
**Contribution:** 3
**Rating:** 4
**Confidence:** 3

**Summary:**

The study explores a key question about modern AI models: does layer-wise dynamics of information processing in transformers reflect real-time processing in human cognition? Built upon the theoretical insights into human cognition in competitor interference, the authors first checked whether mechanistic signatures of competitor interference were presented in models' forward passes. In a following experiment, more features that represent dynamics of forward passes in transformers were extracted and used to predict human performance (accuracy and reaction time) across tasks of competitor interference. Overall, the study supports the predictive power of the hidden dynamics of information in transformers to human processing, revealing the existence of common patterns in cognitive process between model and humans.

**Strengths:**

1. Richness of dynamic metrics. The study proposed a variety of measurements for quantifying both static and dynamic features during the forward passes in transformers, which shows reference value for future research into dynamics processing of AI models.
2. The study tested competitor interference with 20 open-sourced models in 6 domains, providing a robust basis for the discovery.
3. The study addresses a central question to modern AI and cognitive science: can we rely upon AI models to conduct cognitive research, as AI can produce human comparable behavior? The study suggests some level of alignment of real-time processing between humans and AI, providing valuable evidence for future studies in AI-based cognitive models.
4. The result of interaction between model size and predictive power is interesting.

**Weaknesses:**

1. Lack of interpretability. LLMs have shown impressive performance in imitating human behavior through learning from massive human data. However, it doesn't necessarily mean the underlying algorithms learned by LLMs are similar with what humans employ, given the completely different architectures. Thus, implications from similar dynamic internal process may be tricky to apply to human cognitive models.
2. Noisy outcomes from LLMs. As the study used base pretrained LLMs, token activation can be extremely noisy. If we look at the real output of LLMs in the tasks, we might find their terrible handling of the prompts. In this study, the authors used the first token of targeted answers to calculate the log probability of the answer, which may involve a great deal of noise and reduce the reliability of resulted metrics.
3. Lack of statistical results. In study 2 (mainly), some statistics need to be provided when reporting the results that dynamic measures improve prediction of human processing measures (e.g., means and standard errors of Log BF).

**Questions:**

1. In line 132, the authors mentioned 6 dependent variables, while in the following text I just found 5 (Acc, RT, and 3 typing patterns). Do I miss something here?
2. In the analyses of study 2, the reported R-squared is the adjusted or not? If not, adjusted R-squared can be a fair indicator for the predictive power of dynamic IV.
3. Given the great variability among models in their predictive power on human measurements, I wonder how many models show significant predictive power and how many not?

---

### Official Review · Reviewer_sNBc · 2025-10-30

**Soundness:** 3
**Presentation:** 3
**Contribution:** 2
**Rating:** 6
**Confidence:** 4

**Summary:**

In this work, the authors examine whether language models (LMs) offer a useful framework for studying real-time processing strategies in humans. They approach this by introducing novel probabilistic measures to analyze the relationship between layer-wise computational dynamics in transformers and human real-time processing.

First, they show that two of their measures reveal mechanistic signatures of competitor interference in LMs: when tested on vignettes where humans experience decision conflicts, incorrect answers tend to dominate in the initial layers before giving way to correct ones.

Building on this, the authors introduce additional metrics that capture aspects of processing dynamics, such as confidence (absolute and relative), uncertainty, and layer-specific utilities. They then demonstrate that these LM-derived measures, when the models perform the same tasks as humans, reliably predict human processing metrics—including accuracy, reaction time, and time to first keyboard press—while maintaining natural correspondences between the model’s metrics and human data.

Finally, by repeating these analysis across 20 different LMs with different sizes and families, they find that the relationship between model size and predictive performance on human processing measures is task-dependent and not strictly monotonic.

**Strengths:**

The paper is generally very clearly written, with measures and experiments explained and tested reasonably well.
The authors provide a neat way to use LLMs to gain insights into processes underlying human decision making.
The measures derived are intuitive and domain-general, and critically, capture aspects of human real-time processing that are hard to access purely via behavioral experiments.

**Weaknesses:**

Mean predictive performance at least visibly looks very similar same across all measures with the scatter plot, especially in comparison to static measures, in Figure 3a. Maybe the authors should conduct additional statistical analysis comparing the different measures and perhaps show some average measures in the same plot for reference. The R2 values (mean over participants) over model sizes shown in Figure 3b seem quite low. Can we make any serious claims based on this?

The focus of this work correlational analysis and is missing any causal tests. For example, it would interesting to intervene on these models based on some of these measures to see if they lead to different final outcomes. This could also shed light on how one can design human experiments to probe the utility of these measures.

Mechanistic insights into what aspects of human processing do of these measures successfully capture are missing: why are they predictive of behavior, how are these findings related to previous findings, and how can these findings be used to design human experiments to probe the discovered processes.

**Questions:**

What are some measures of processing in networks are actually not predictive of human behavior as control?

Are there additional baselines from the cognitive science literature that can predict RTs, accuracy, etc and how do these new measures compare against them in terms of predictive power?

What is the intuition behind why larger models are bad? is it because deeper models have more processing times than needed for such simple tasks and might have converged onto the correct answer early on?

---

### Official Review · Reviewer_wAvb · 2025-10-31

**Soundness:** 2
**Presentation:** 2
**Contribution:** 2
**Rating:** 4
**Confidence:** 3

**Summary:**

The authors of this paper attempt a bold interdisciplinary move: using the internal, layer-by-layer processing dynamics of Transformer models to predict human cognitive load. In other words, they want to see if what happens inside a neural network’s "mind" can tell us something nontrivial about real human minds as they process information in real time.

The strongest part of this paper is its ambition. The authors are effectively proposing a new way to use AI interpretability: as a speculative mirror for human cognitive processes. The paper is full of intriguing hints and it’s refreshing to see this kind of conceptual crossover.

That said, I have significant reservations about the robustness of the claims. The authors do not appear to report any p-values or formal tests of statistical significance, and I do not think that the authors want to die on the hill of having to defend their methodological choice of "improving fit of a linear model". In my view, this is like playing tennis with the net down. Without a clear demonstration that their added "dynamic" measures improve predictive accuracy in a statistically significant way, the argument is left floating in suggestive but ungrounded territory. In short, they haven’t shown that these insights are much more than an intriguing curiosity to any epistemic standard I would bet money on.

In addition, there’s the matter of interpretation. The authors seem to imply that these layer-by-layer dynamics tell us something fundamental about human cognitive stages. But without a stronger theoretical or empirical grounding, this is more of a speculative leap than a demonstrated finding. The risk is that the reader is left to imagine a deeper connection than the evidence actually supports.

In sum, I appreciate the authors’ creative attempt to marry AI interpretability with cognitive science. But for this to be more than a curiosity, they either need to do the statistical legwork, or reposition their paper as the fishing expedition it is. Without that, the paper reads like a promising hypothesis in search of a robust demonstration.

**Strengths:**

- Ambitious cross-disciplinary attempt to connect model layer dynamics with human cognitive timing.
- Uses multiple human datasets and multiple model families.
- Introduces simple, interpretable dynamic measures (COM, TTD) of layer-wise preference shifts.

**Weaknesses:**

- No formally evaluable claims of significance; claims rest on descriptive trends.
- “Human-like processing” inference overreaches the data.
- Conceptual link between forward-pass depth and cognitive time lacks theoretical grounding.
- Linear regressions on behavioral metrics may overfit noise; unclear robustness.

**Questions:**

- Are COM and TTD stable across random seeds or dependent on layer depth only?
- How do the results compare to recurrent or diffusion-based models with different architectures? i.e. could it rather be a matter of the training data?
- Could dynamic measures be predictive of human reaction times on unseen tasks, not just correlated?

---

### Note · Authors · 2025-11-24

I have read and agree with the venue's withdrawal policy on behalf of myself and my co-authors.